# Detection Method and Common Characteristics of Waste Solvent from Semiconductor Industry

**DOI:** 10.3390/molecules28165992

**Published:** 2023-08-10

**Authors:** Jinjuan Ni, Qing Zhang, Xianglin Zhang, Zhilong Sun, Dali Bao

**Affiliations:** 1School of Resource and Environmental Engineering, Hefei University of Technology, Hefei 230009, China; 15955116486@126.com (J.N.); 2019180191@mail.hfut.edu.cn (Z.S.); 2019010072@mail.hfut.edu.cn (D.B.); 2CAS Key Laboratory of Crust-Mantle Materials and Environment, School of Earth and Space Sciences, University of Science and Technology of China, Hefei 230026, China; 3School of Resource and Environmental Engineering, Anhui Water Resources Hydropower Technical College, Hefei 231603, China

**Keywords:** TFT-LCD, waste organic solvent, common characteristics, unified dtection method

## Abstract

The recycling of organic solvents is a widely discussed topic. The waste organic solvents from thin-film-transistor liquid-crystal display (TFT-LCD) production is characterized by large quantities, multiple types, and complex compositions. Thus, the unified and compatible component analysis method is important for studying the recovery of waste organic solvents. In our work, based on the study of existing analytical methods, we designed a compatible method for the analysis of moisture using Karl Fischer analysis, for the analysis of organic compounds using gas chromatography, and for the analysis of the photoresist and other solids by evaporation. These were specific methods for analyzing the components of near-total formulation thin-film-transistor liquid-crystal display waste organic solvent. The organic matter content was analyzed via gas chromatography with a CP-Sil8CB column and flame ionization detector. The initial temperature of the column was 90 °C and the holding time was 1 min. The heating rate was 30 °C/min. The temperature was raised to 270 °C for 7 min. The internal standard method and the external standard method were used to determine the content of the main components of organic compounds. The relative standard deviation of the analytical results was 1.14~2.93%, 1.21~4.74% and 0.61%, respectively. The analytical results had good accuracy, but the external standard method was better; the recoveries were 99.76~107.60%, 95.86~107.70%, and 95.23~96.88%, respectively. Based on the composition analysis, the composition rule of the waste organic solvent was summarized. Through the exploration of the effect of the waste solvent, the common characteristics of the waste solvent were obtained. This study provides a good strategy and an optimized method for improving the efficiency of organic solvent recovery.

## 1. Introduction

Currently, the manufacturing of display and semiconductor materials is one of the most important and largest industries in the world. Thin-film-transistor liquid-crystal displays (TFT-LCD), which are a mature technology, have become the main product in the semiconductor industry, and are widely used by people because these displays are lighter, thinner, have a lower power consumption, and cause less harm to the environment [1,2]. However, the rapid development of semiconductors has increased the amount of organic waste solvent from screen production [3,4], which increases the production cost and the risk of environmental damage. The recovery of the organic solvents used in TFT-LCD production has gradually become a much-discussed topic, which has led to extensive research against the backdrop of the need for developing a low-carbon economy [5,6,7]. Basic research on the efficient recovery of waste solvents includes the study of the analysis method of waste solvents. However, at present, there very few studies that have reported on the analysis of waste organic solvent components. There are many waste organic solvents in TFT-LCD production, such as stripper, thinner, and waste *N*-Methyl-2-pyrrolidone (NMP), and each waste solvent usually has multiple formulations, which are generally composed of organic matter, water, and a small amount of photoresist [7,8,9]. It is particularly important to validate an accurate, compatible, and unified method for analyzing each component of the liquid waste. At present, the moisture content of the waste organic solvent is mostly determined using a moisture meter [10,11]. An analysis of the content of photoresist and other solids generally involves the determination of the solid content of coatings using the evaporation method. There are many studies and applications of the use of gas chromatography to analyze organic solvent components [12,13,14,15,16,17,18]. For example, Ulusoy et al. used gas chromatography to determine the content of organic solvents such as ethanol and methanol in blood. Tatebe et al. used gas chromatography (GC) to analyze organic solvent residues in thickeners. However, few studies have been conducted on the use of GC to analyze waste organic solvents from TFT-LCD production. Regarding the important issue of organic solvent recovery, the application of gas chromatography to analyze waste organic solvent from TFT-LCD production is worthy of further study. It is important to establish a compatible and unified method for analyzing the water, principal components of the organic solvent, and the photoresist and other solids in the recovery of organic solvents used for TFT-LCD production which, in turn, can improve the analysis efficiency of waste liquids and reduce the analysis cost and management cost. Developing a method of analyzing the waste solvent composition is crucial for improving the sustainability of this industrial process [5,19,20,21].

In this study, the recovered waste organic solvent was taken as the research object, and the Karl Fischer water analysis method was used to analyze the water content of the waste solvent [10,22,23,24]. Multiple parallel experiments and standard substance recovery experiments were set up to validate the accuracy of the water analysis method regarding the water content detection of the waste organic solvent from TFT-LCD production [25,26,27]. The organic solvents in the waste solvent were collected via evaporation and decompression distillation, and the residues, which included the photoresist and other solids in the waste organic solvents, were weighed using the weighing method. The application of gas chromatography in TFT-LCD waste organic solvent was discussed based on the material characteristics of different components, the compatible program analysis method was established, and the accuracy of the compatible method was verified using an internal standard method and external standard method [28,29,30,31]. The aim was to explore and establish a unified analysis method that is applicable to analyzing the contents of the water, organic matter, and photoresist and other solids in the waste solvent, including strippers, thinners, and NMP, from TFT-LCD production. From the perspective of the industry, this paper combined the advantages of existing analytical methods in component analysis to develop a compatible analysis method for waste organic solvent in the semiconductor industry, so as to reduce the analysis cost and improve the efficiency of waste solvent analysis and management. The common characteristics of waste solvents were studied based on the compatibility analysis method. These studies provide accurate basic information for the development of a waste solvent recovery process in the semiconductor industry, and further improve the recovery rate and efficiency of solvent, which is also a strong guarantee for the development of a low-carbon economy.

## 2. Results and Discussion

### 2.1. Moisture Detection

#### 2.1.1. Repetitive Experiment

The reproducibility of the present method was investigated via a sextuplicate de-termination of the moisture content of the waste solvent samples from three classes of five formulations under the preferred conditions. The relative standard deviation (RSD) of the water determination was less than 5.00%. The results are presented in Table 1.

#### 2.1.2. Standard-Added Recovery Rate Experiment

To evaluate the accuracy of the proposed method, experiments with a labeled recovery rate were performed. Water Standard 10.0 (Honeywell, Selzer, Germany) (1%, 2%, and 3%) was added to the five waste organic solvent samples, and the water contents of the labeled samples were determined using the method described in Section 3.3.1, with six replicates, to determine the recovery rate of the measurement method. The results are presented in Table 2. It can be seen from Table 2 that the recovery rates of the moisture contents of the five waste organic solvents were 96.63~101.20%, which shows that this method has sufficient accuracy and reliability for the determination of the moisture content of waste organic solvents from TFT-LCD production.

### 2.2. Detection of Organic Matter Content

#### 2.2.1. Detection of Main Organic Content in Waste Stripper and Thinner

(1) To evaluate the accuracy of the method, the internal standard method and external standard method experiments were performed separately to compare the precision of the experimental results. The standard curves for NMF, MDG, BDG, PGMEA, and PGME are shown in Figure 1. The experiment was investigated using sextuplicate determination, and the mean value, standard deviation (S), and relative standard deviation (RSD) were calculated (Table 3 and Table 4).

The results in Table 3 show that the RSD for the waste stripper solvent was 1.14–1.90% using the external standard method, and the RSD was 1.25~2.93% using the internal standard method. The results in Table 4 show that the RSD for the waste thinner solvent was 1.21~3.14% using the external standard method, and the RSD was 1.36%~4.74% using the internal standard method. Based on a comparison of these results, the measurement results of the external and internal standard methods have little difference in terms of accuracy, and both methods have a good reproducibility for the sample detection results. However, the external standard method does not require the use of an external standard, so the operation is simpler, and the error is relatively low, making it more suitable for the determination of the organic matter content of waste stripper and waste thinner solvents.

(2) Based on the external standard method, we conducted an experiment to further verify the reliability of the method. The results and recovery are presented in Table 5.

The standard recovery rates of the organic matter content in the two formulations of waste stripper solvent were 99.76~107.60%, and those of the two formulations of waste thinner solvent were 95.86~107.70%, demonstrating that the proposed method is sufficiently accurate and robust regarding the determination of the organic matter content of waste stripper solvent.

#### 2.2.2. Detection of Main Organic Content of Waste NMP Solvent

The reproducibility of the proposed method was investigated via sextuplicate determination of the NMP in the waste NMP solvents under the preferred conditions. The mean value, standard deviation (S), and relative standard deviation (RSD) were also calculated. The specific results are presented in Table 6.

The results presented in Table 7 show that the standard recovery rate of the NMP in the waste NMP solvent was 95.23–96.88%, demonstrating that the proposed method is sufficiently accurate and robust for the determination of the NMP content of waste NMP solvent. Figure 2 shows the GC analysis chromatogram of the waste solvents.

### 2.3. Detection of the Photoresist and Other Solids

The sextuplicate determination of the photoresist and other solids in the waste solvents under the preferred conditions was conducted. The mean value, standard deviation (S), and relative standard deviation (RSD) were also calculated. The specific results are presented in Table 8.

The results in Table 8 show that the RSD of the analysis results was <4%, demonstrating that the distillation method of determining the solid impurities in the waste liquid is accurate and reliable.

### 2.4. Analysis of Waste Solvent Composition and Common Characteristics

In the early stage of the development of the flat panel display industry, there were many types and formulations of electronic chemicals used, and the waste solvent components were different, which was not conducive to the organic solvent recovery units [32,33,34,35,36,37]. This study started from the composition analysis of waste solvent, established a unified and efficient component analysis method, and summarized the composition analysis results of waste solvent. Table 9 is the summary of the analysis results of the six waste solvents of TFT-LCD using the approach established above.

As shown in Table 10, based on the above conclusions and the function of the role of electronic chemicals in the screen-making process, the common characteristics of such waste solvent were found out. This laid a good foundation for stable research on the compatible recovery process of electronic recovery products that met the reuse standard, and provided basic research for solving the problem that the existing recovery process system could only match the recovery of one formula of waste solvent.

## 3. Materials and Methods

### 3.1. Chemicals and Reagents

The Karl Fischer Reagent was purchased from the Tianjin Comio Chemical Reagent Co., Ltd., Tianjin, China. The *N*-methylformamide (NMF; GR), diethylene glycol methyl ether (MDG; GR), diethylene glycol monobutyl ether (BDG; GR), *N*-methylpyrrolilione (NMP; GR), propylene glycol methyl ether acetate (PGMEA; GR), and propylene glycol methyl ether (PGME; GR) were guaranteed reagent grade and were obtained from the Sinopharm Group Chemical Reagent Co., Ltd., Shanghai, China. The solution was prepared using deionized (DI) water (18.22 MΩ/cm). The waste liquid of organic solvent used in the experiment came from the top three panel production plants in the world. 

### 3.2. Experimental Device

A moisture meter (Metler V20 equipped with a double needle platinum electrode); gas chromatograph (Agilent GC7890A equipped with a flame ionization detector (FID), automatic sample injection, and 10-μL microinjection); baking box (Shanghai Keheng Industrial Development Co., Ltd., Shanghai, China); and electronic balance (Shanghai Shangping Instrument Co., Ltd., Shanghai, China, FA2004, precision 0.0001 mg) were used in the experiments. The vacuum pressure distillation experiments were carried out in 250 mL three-mouth glass flasks, which were placed in an oil bath with a magnetic stirrer. The material was evaporated under pressure in the flask, the system was a closed loop, and the gas phase was condensed at room temperature. The condensate was collected in a 250 mL tapered flask, and the exhaust gas was absorbed by deionized water. A thermocouple was used to control the reaction temperature, with an accuracy of ±0.2 °C. After the experiment was completed, a sample was collected after the in-bottle temperature was thoroughly cooled.

### 3.3. Experimental Methods

#### 3.3.1. Determination of Moisture Content

(1) Determination conditions

The starting maximum drift value was 25 μL/min; the electrode was a polarized double platinum needle electrode; and the polarization current was 24 μA. The titration end point was 120 mv; the drift value was 50 μL/min; the maximum hydration rate was 0 mL/min; the minimum hydration rate was 100 μL/min; and the mixing time was 15 s.

(2) Determination methods

After calibrating the moisture meter using pure water, 0.5–1.0 g of sample was added to the titration cup (accurate to 0.0001 g). Most of the organic compounds in the sample that needed to be tested had a certain volatility; therefore, a syringe was used to absorb the sample during testing to avoid sample volatilization and water absorption. The weight difference method was used to calculate the mass of the sample added. The mixture was stirred for 60 s and titrated. At the end of the titration, the instrument automatically calculated the moisture content of the sample and displayed it on the screen in the form of a percentage. The experimental results were recorded, and the experiment was repeated.

#### 3.3.2. Determination of the Organic Matter Content

The waste organic solvent from TFT-LCD production contains many types of complex components, so the GC conditions must be carefully selected [38,39].

(1) Determination conditions

① Sampling method: An automatic sample injection device (equipped with a 10 μL sample injection needle) was selected to reduce the error of the sample injection amount because the sample detection method includes the external standard method.

The contents of the tested components in the sample were high, so a shunt in-take was adopted to optimize the chromatography peak type and avoid tail drag of the chromatography peak. The shunt was set to 150:1 and the sample intake was set to 1 μL.

② Chromatographic column: The analyte contained amines. A capillary column (Agilent CP-Sil8CB) fixed with 5% alkali deactivation technology in the stationary phase was used [40,41,42,43,44,45]. The column had a flow velocity of 1 mL/min and dimensions of 30 m × 320 μm × 1.0 μm, and the maximum temperature could be set to 325 °C, which ensured adequate evaporation of the analyte. The content of the other organic impurities and the degree of separation were small, and thus, the column length was 30 m.

③ Detector: The highest temperature of the column was 325 °C, and the detector temperature was less than 325 °C. Considering the boiling point of each component, the FID temperature was set to 300 °C. The hydrogen flow was 40 mL/min; the air flow was 400 mL/min; and the nitrogen flow was 30 mL/min.

④ The inlet conditions: The inlet temperature was set to 270 °C according to the highest boiling point of the tested components. The flow rate was 156 mL/min; and the cushion purge flow rate was 5 mL/min.

⑤ Column box condition: According to the boiling point of each component to be tested, the initial temperature was set to 90 °C and the holding time was 1 min. The heating rate was 30 °C/min; the temperature was increased to 270 °C; and the holding time was 7 min.

(2) Determination method

The gas chromatograph was prepared, 1–1.5 mL of sample was placed in the automatic sample bottle, and the detection was started after the measurement method was set. The instrument sequence table was opened to edit the sample name, detection repetition times, and other details. The process started automatically, and the experimental results were recorded after the operation. The external standard method and the internal standard method were used to determine the organic matter content of the waste stripper and thinner solvents, respectively. The labeling experiment was set up. For the NMP waste solvent, since the NMP content reached 95%, the direct sample entry analysis and a calibration experiment were used to calculate the percentage content. 

External standard method is a quantitative method using the pure product of the component to be measured as the reference material, and comparing the response signal (peak area) of the component to be measured in the reference material and the sample. This method is simple to operate. Internal standard method is a quantitative analysis method that selects the pure substance not contained in the sample as the internal standard, and compares the response signal (peak area) of the component to be measured in the internal standard and the sample. Both methods can accurately quantify organic components. The accuracy of the external standard method is affected by the repeatability of sample injection and the stability of experimental conditions. In this study, the determination of samples was set parallel for 6 times, and the experimental conditions were set strictly in accordance with the determination conditions (1) in 3.3.2. Internal standard method has some problems such as difficult selection of internal standard object and troublesome configuration. In this study, according to the main components of waste stripper and thinner, NMP was selected as the internal standard because there was no NMP in the sample to be tested, and NMP could be miscible with the solution to be tested, without chemical reaction with the components in the sample to be tested, and could be completely separated from the sample to be tested. The above two methods were used to determine, and the average value, RSD, and standard deviation of the measured results were compared in order to select the suitable method.

(3) Allocation of the standard curve solution using the external standard method

① Configuration of NMF + MDG, NMF + BDG standard solution: Guaranteed rea-gent was used to create solutions with NMF contents of 20%, 30%, 40%, 50%, and 60%, with corresponding MDG contents of 80%, 70%, 60%, 50%, and 40%. Guaranteed reagent was used to create solutions with NMF contents of 20%, 30%, 40%, 50%, and 60% with corresponding BDG contents of 80%, 70%, 60%, 50%, and 40%. The NMF percentage content, MDG percentage content, and the peak areas were calculated separately, and the NMF percentage content and BDG percent-age content were calculated using the same method.

② Configuration of the PGMEA external standard solution: Standard solutions with PGMEA contents of 20%, 35%, 50%, 65%, and 80% were created. The standard working curve of the relationship between the PGMEA percentage content, and the peak area was obtained.

③ Configuration of PGME external standard solution: A standard solution was pre-pared with PGME contents of 20%, 35%, 50%, 65%, and 80%. The standard working curve of the relationship between the PGME percentage content, and the peak area was obtained.

(4) Allocation of the standard curve solution using the internal standard method:

We took the configured external standard solution, added NMP pure solution, according to the mass ratio of 4:1 for each standard solution, and configured the corresponding concentration of internal standard solution.

### 3.4. Determination of the Photoresist and Other Solids

#### 3.4.1. Method Principle

The photoresist and other solids in waste solvents are non-volatile substances, so the method for determining the solid content of coatings was used. The waste organic solvent was distilled under reduced pressure, and all of the organic solvent substances were evaporated; the remaining substances were the content of the photoresist and other solids [46].

#### 3.4.2. Determination Methods

A 250 mL tapered flask was heated in a 105 ± 2 °C oven for 30 min, removed, placed in the dryer for 30 min, and weighed after the tapered flask had cooled to room temperature. Then, 10–20 g of uniform organic waste solvent was placed in a conical bottle, and a pressure-reducing distillation system was constructed, which included a heating system, coolant passage, and condensate collection bottle. The selected experimental pressure was 16.66 kPa. The decompression distillation was started. The temperature was gradually increased to 180 °C within 1–2 h. When the level in the condensate collection bottle no longer changed and the photoresist in the tapered flask had been evaporated, the conical bottle was heated in a 105 ± 2 °C oven for 30 min, removed, placed in a dryer for 30 min, cooled, and weighed. The bottle was heated in a 105 ± 2 °C oven for an additional 30 min, removed, placed in a dryer for 30 min, and weighed. This operation was repeated until the difference between two consecutive weights was less than 0.1 g (all the weighs were accurate to 0.1 g). Parallel samples were tested to calculate the content of photoresist and other solids in the waste according to the weight difference.

## 4. Conclusions

In this study, the composition, co-physicochemical characteristics, and available components of near-full-formula organic solvent wastewater for thin-film-transistor liquid-crystal displays (TFT-LCD) were studied, creating a compatible analysis method for the analysis of moisture using Karl Fischer analysis, for the analysis of organic compounds using gas chromatography, and for the analysis of the photoresist and other solids by evaporation. These are specific methods for analyzing the components of near-total formulation TFT-LCD waste organic solvent. In this paper, a compatible analysis method of waste solvent compositions of organic solvent for TFT-LCD was developed using the existing analytical methods. The new method avoids switching in the analysis of different waste solvent compositions and is more beneficial to the management of waste solvent. Based on the composition analysis, the composition rule of the waste organic solvent was summarized. TFT-LCD waste organic solvents usually contain more than 90% of the main organic components, less than 5% of the water, and less than 5% of the photoresist and other solids. Through the exploration of the effect of the waste solvent, the common characteristics of the waste solvent were obtained. The main compositions of the stripper usually contain ether and amine organic solvents, and the thinner mainly contain propylene glycol methyl ether acetate and propylene glycol methyl ether, both of which, along with *N*-methylpyrrolidone, have a strong ability to dissolve photoresist and other polymer materials. This study provides a good strategy and an optimized method for improving the efficiency of organic solvent recovery. 

Based on this research, a compatible recovery process for waste organic solvents used in the semiconductor industry will be developed. At present, the research method is only for the detection of waste organic solvents used in the semiconductor industry, and the idea of considering the industry as a whole can be applied to other fields. 

## Figures and Tables

**Figure 1 molecules-28-05992-f001:**
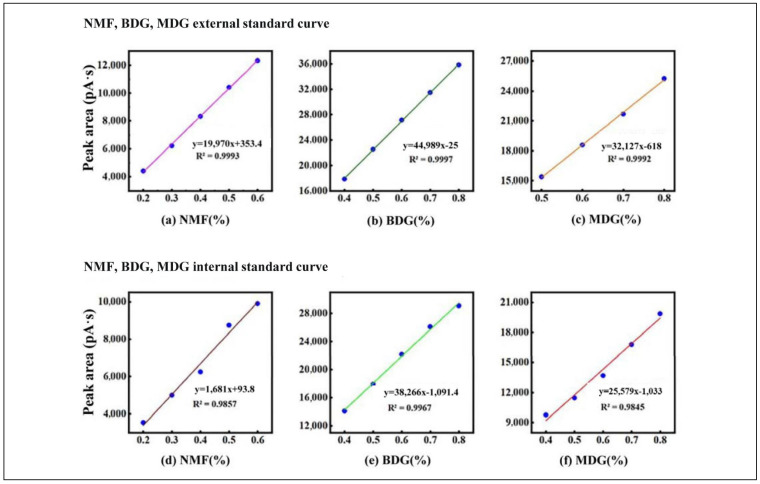
Standard curves and GC spectra of the main organic components of waste stripper and thinner by internal and external standard methods.

**Figure 2 molecules-28-05992-f002:**
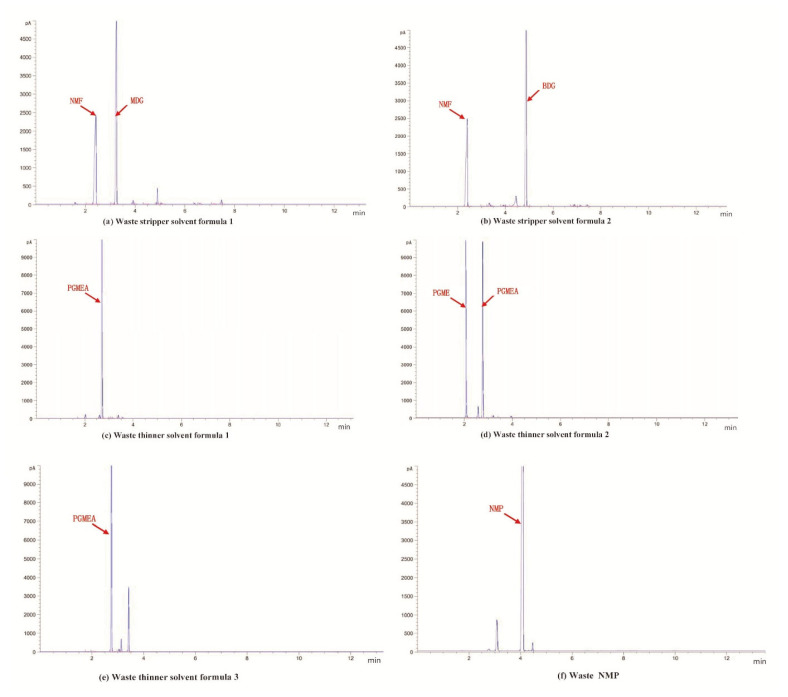
The GC analysis chromatogram of the waste solvents.

**Table 1 molecules-28-05992-t001:** Moisture determination results.

Parallel Samples (*n*)	Waste Stripper Solvent Formula 1 (NMF + MDG) (%)	Waste Stripper Solvent Formula 2 (NMF + BDG) (%)	Waste Thinner Solvent Formula 1 (PGMEA) (%)	Waste Thinner Solvent Formula 2 (PGMEA + PGME) (%)	Waste NMP Solvent (%)
1	3.0187	2.9879	1.7212	1.8854	4.2278
2	2.9977	3.2312	1.8120	1.7845	4.3215
3	3.2123	3.1132	1.6950	1.8252	4.1259
4	3.1822	3.0182	1.7763	1.7545	4.4458
5	2.8990	2.8451	1.7223	1.7832	4.4812
6	3.1563	3.1211	1.8121	1.7702	4.3678
Standard deviation	0.1241	0.1332	0.0505	0.0478	0.1341
Average value (%)	3.0777	3.0528	1.7565	1.8005	4.3283
RSD (%)	4.0329	4.3627	2.8762	2.6529	3.0972

**Table 2 molecules-28-05992-t002:** Experimental results of water labeling (n = 6).

Sample	Water Background Value (%)	Increase in the Amount of Standard Water Sample (%)	Measured Value (%)	Percent Recovery (%)
Waste stripper solvent formula 1 (NMF + MDG)	3.0777	1.0000	4.0652	98.75
		2.0000	5.1017	101.20
		3.0000	6.0513	99.12
Waste stripper solvent formula 2 (NMF + BDG)	3.0528	1.0000	4.0204	96.76
		2.0000	4.9970	97.21
		3.0000	5.9790	97.54
Waste thinner solvent formula 1 (PGMEA)	1.7565	1.0000	2.7385	98.20
		2.0000	3.6921	96.78
		3.0000	4.6554	96.63
Waste thinner solvent formula 2 (PGMEA + PGME)	1.8005	1.0000	2.7668	96.63
		2.0000	3.7331	96.63
		3.0000	4.6994	96.63
Waste NMP solvent	4.3283	1.0000	5.3150	98.67
		2.0000	6.3087	99.02
		3.0000	7.2779	98.32

**Table 3 molecules-28-05992-t003:** Analysis results of organic components in waste stripper solvent.

Parallel (*n*)	External Standard Method				Internal Standard Method			
	Formula 1 (NMF + MDG)		Formula 2 (NMF + BDG)		Formula 1 (NMF + MDG)		Formula 2 (NMF + BDG)	
	NMF (%)	MDG (%)	NMF (%)	BDG (%)	NMF (%)	MDG (%)	NMF (%)	BDG (%)
1	31.89	61.46	33.17	64.17	32.23	62.08	33.22	63.22
2	31.98	61.45	33.56	62.45	34.55	64.12	34.77	64.55
3	32.03	62.76	34.12	63.11	34.76	63.05	33.67	62.17
4	32.14	62.86	33.34	64.75	32.89	63.87	34.22	64.88
5	33.50	62.33	34.07	62.83	33.41	62.44	33.45	65.12
6	32.00	61.25	33.67	63.37	33.12	63.01	34.01	65.44
Standard deviation	0.61	0.72	0.38	0.86	0.98	0.79	0.56	1.27
Average value (%)	32.26	62.02	33.66	63.45	33.49	63.10	33.89	64.23
RSD (%)	1.90	1.16	1.14	1.36	2.93	1.25	1.66	1.97

**Table 4 molecules-28-05992-t004:** Analysis results of organic components in waste thinner solvent.

Parallel (*n*)	External Standard Method			Internal Standard Method		
	Formula 1 (PGMEA)	Formula 2 (PGMEA + PGME)		Formula 1 (PGMEA)	Formula 2 (PGMEA + PGME)	
	PGMEA (%)	PGMEA (%)	PGME (%)	PGMEA (%)	PGMEA (%)	PGME (%)
1	75.55	53.23	22.65	75.22	55.61	21.34
2	78.23	54.34	22.47	75.89	52.37	22.67
3	76.37	55.86	21.67	74.28	58.24	22.1
4	77.12	53.75	23.01	77.05	55.31	22.92
5	76.34	54.1	21.11	75.98	56.73	23.16
6	76.21	54.82	22.05	76.84	54.28	24.55
Standard deviation	0.93	0.91	0.69	1.03	2.02	1.08
Average value (%)	76.64	54.35	22.16	75.88	55.42	22.79
RSD (%)	1.21	1.68	3.14	1.36	3.64	4.74

**Table 5 molecules-28-05992-t005:** Experimental results of organic component labeling (n = 6).

Sample		Background Values (%)	Amount Added (%)	Measured Value (%)	Percent Recovery (%)
Waste stripper solvent formula 1 (NMG + MDG)	NMF mark	32.26	1	34.00	102.31
			2	33.97	99.12
			3	37.12	105.79
	MDG mark	61.83	1	62.68	99.76
			2	66.82	104.83
			3	68.55	106.01
Waste stripper solvent formula 2 (NMG + BDG)	NMF mark	33.66	1	36.75	106.23
			2	37.08	104.23
			3	39.21	107.60
	BDG mark	63.45	1	65.23	101.23
			2	68.55	104.89
			3	67.15	101.11
Waste thinner solvent formula 1 (PGMEA)	PGMEA mark	75.88	2	75.40	96.74
			5	80.21	99.12
			10	82.74	95.86
Waste thinner solvent formula 2 (PGMEA + PGME)	PGMEA mark	54.35	2	59.67	106.10
			5	62.35	105.52
			10	68.54	107.70
	PGME mark	22.16	2	25.78	107.30
			5	28.50	106.00
			10	32.41	101.11

**Table 6 molecules-28-05992-t006:** Analysis results of NMP in waste NMP solvent.

Parallel (*n*)	1	2	3	4	5	6	Standard Deviation	Average Value (%)	RSD (%)
NMP (%)	94.21	95.13	93.79	94.58	95.02	95.25	0.58	94.66	0.61

**Table 7 molecules-28-05992-t007:** Experimental results of NMP (n = 6).

Background Value (%)	Amount Added (%)	Measured Value (%)	Percent Recovery (%)
94.66	1	92.61388	96.78
	2	92.14662	95.23
	3	94.70855	96.88

**Table 8 molecules-28-05992-t008:** Results of the determination of the photoresist and other solids.

Parallel (*n*)	Waste Stripper Solvent Formula 1 (NMF + MDG)	Waste Stripper Solvent Formula 2 (NMF + BDG)	Waste Thinner Solvent Formula 1 (PGMEA)	Waste Thinner Solvent Formula 2 (PGMEA + PGME)	Waste NMP Solvent (%)
1	1.5400	1.8500	1.4100	3.4200	1.1100
2	1.4900	1.8100	1.4300	3.5200	1.0700
3	1.5800	1.7000	1.4500	3.5500	1.1400
4	1.5500	1.8800	1.3900	3.6100	1.0900
5	1.5000	1.7900	1.3200	3.4900	1.1700
6	1.4700	1.7400	1.4900	3.7300	1.1300
Standard deviation	0.0417	0.0672	0.0579	0.1071	0.0360
Average value (%)	1.5217	1.7950	1.4150	3.5533	1.1183
RSD (%)	2.7387	3.7413	4.0904	3.0136	3.2199

**Table 9 molecules-28-05992-t009:** The analysis results of TFT-LCD waste organic solvent.

Waste Solvent	Process Source	Composition of the Waste
Waste stripper	Array	NMF + MDG > 90%, water < 5%, the photoresist and other solids < 5%
	Array	NMF + BDG > 90%, water < 5%, the photoresist and other solids < 5%
Waste thinner	Array	PGMEA > 98%, water < 2%, the photoresist and other solids < 5%
	Array/CF	PGME + PGMEA > 98%, the photoresist and other solids < 5%, water < 2% (PGME:PGMEA3:7)
	CF	(PGME + PGMEA + methyl 3-methoxypropionate + Acetic Acid 3-Methoxybutyl Ester + Cyclohexene et al.) > 98%(Of these, PGMEA > 75%), water < 2%, the photoresist and other solids < 5%
Waste NMP	Cell	NMP > 90%, water < 5%, the photoresist and other solids < 5%

**Table 10 molecules-28-05992-t010:** Common characteristics of organic solvents used for TFT-LCD.

Waste Solvent	Composition	Commonness	Function Introduction
Waste stripper	NMF, MDG, BDG, the photoresist and other solids, water	Amines and ethers	Ethers make the photoresist foam expansion easy to peel off; amines infiltrate, peel, and dissolve the photoresist with the substrate, the effect of the amine-dissolved photoresist is mainly reflected in the product of the crosslinking reaction of the dissolved photoresist after exposure and promotes the occurrence of the dissolved crosslinking reaction.
Waste thinner	PGMEA, PGME, the photoresist and other solids, water	PGMEA, PGME	PGMEA and PGME dissolve the photoresist more evenly on the glass substrate. The molecules of PGMEA have both ether bonds and groups, which form an ester structure and contain an alkyl group. In the same molecule, both the nonpolar part and the polar part have strong dissolution capacity.
Waste NMP	NMP, the photoresist and other solids, water	Polar nonproton delivery solvent	NMP has a excellent dissolution ability, stable chemical performance, small toxicity, and strong biodegradation ability.

## Data Availability

The data presented in this study are available within the article.

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
