# Peer review of "Detection Method and Common Characteristics of Waste Solvent from Semiconductor Industry"

_molecules, 2023, doi:10.3390/molecules28165992_

Round 1

Reviewer 1 Report

Authors have presented an article on “Detection method and common characteristics of waste solvent from semiconductor industry” The manuscript in the current state can be accepted for publication after minor modifications.

Comments

1. The quality of the used images need to be improved so that the inside details can be easily read.

2. Language can be improved in all the sections.

3. Authors must explain about re-producibility of results of experiment.

4. How can this technology improve the humankind should also be added.

After these changes, the article can be considered for publication.

Authors have presented an article on “Detection method and common characteristics of waste solvent from semiconductor industry” The manuscript in the current state can be accepted for publication after minor modifications.

Comments

1. The quality of the used images need to be improved so that the inside details can be easily read.

2. Language can be improved in all the sections.

3. Authors must explain about re-producibility of results of experiment.

4. How can this technology improve the humankind should also be added.

After these changes, the article can be considered for publication.

Reviewer 2 Report

The authors reported a general strategy for analysis of waste solvent from semiconductor industry. References are up to date . The paper is well written and organized. Thanks.   However, many recommendations should be improved.

1-     Novelty and merits for the new approach should be more highlighted.

2-     Why did the authors select NMP as the internal standard?

3-     Add proper reference for 3.4. Determination of the photoresist and other solids.

4-     Conclusion should be written with full names [PGMEA and PGME, thinner and NMP]

5-     Chemical structures for organic components in waste stripper and thinner solvents should be displayed as figure 1.

6-     In 2) Determination method; full details, descriptions, and differences should be stated for the external standard method and the internal standard method (NMP was used as the internal standard).

7-     Abbreviation list is strongly recommended

8-     future plan and study limitation should be highlighted

Best wishes

Reviewer 3 Report

The manuscript, Detection method and common characteristics of waste solvent from semiconductor industry, authored by Ni et. al studies the detection and characterization of waste solvents from the semiconductor industry. There are only 2 minor comments to be made.

1. It is unclear what is the waste source. The authors can specify this in the experimental section.

2. The authors did should be done with the recovered waste solvent. The authors should add a paragraph in the introduction section or in the discussion section.
